# Subjective Functional Difficulties and Subjective Cognitive Decline in Older-Age Adults: Moderation by Age Cohorts and Mediation by Mentally Unhealthy Days

**DOI:** 10.3390/ijerph20021606

**Published:** 2023-01-16

**Authors:** Renata Komalasari, Elias Mpofu, Gayle Prybutok, Stanley Ingman

**Affiliations:** 1Department of Rehabilitation and Health Services, University of North Texas, Denton, TX 76201, USA; 2School of Health Sciences, University of Sydney, Camperdown, NSW 2050, Australia; 3Educational Psychology, Faculty of Education, University of Johannesburg, Johannesburg P.O. Box 524, South Africa

**Keywords:** affective reactivity, chronic stress theory of aging, depressive symptoms, mentally unhealthy days, subjective functional difficulties, subjective cognitive decline

## Abstract

**Background and Objectives**: Despite the expected positive association between subjective functional difficulties (SFD) and subjective cognitive decline (SCD), their mediation by mentally unhealthy days (MUDs) is under-studied. Moreover, few SCD studies have examined affective reactivity to functional difficulties by age cohort. This study examined the mediation effect of MUDs on the association between SFD and SCD by age cohorts’ moderation among older adults. **Methods:** Using a cross-sectional design approach, we used the 2019 BRFSS dataset on 13,377 older adults aged 65 to ≥80 (44% males and 56% females). **Results:** The bias-corrected percentile bootstrap with 5000 samplings revealed that MUDs partially mediate the relationship between SFD and SCD (14.12% mediation effect), controlling depressive symptoms, education, income levels, and race. Age cohorts moderated the relationship between SFD and SCD and between SFD and SCD but not between MUDs and SCD. The predictive effects of the path from SFD to MUDs and from MUDs to SCD were stronger in the younger-old (65–69) than the middle-old (70–79) and oldest-old (≥80) age cohorts. Worse SCD was associated with being Asian, in female older adults, and in those with lower education and income levels. **Conclusions:** These findings extend the chronic stress theory predictions that accentuated emotional vulnerability to subjective functional difficulties may magnify SCD, particularly in the younger-old age group. By implication, preventive SCD interventions should seek to support younger-old adults in their activities of daily life for successful aging transitioning into older-age cohorts.

## 1. Introduction

Older adults who self-report difficulties in activities of daily living (ADL) tend to report subjective cognitive decline (SCD) [1,2]. SCD is when an individual expresses concerns about an increasing need for support in mental activities the person was capable of previously [3]. A person with SCD might have a subjective memory complaint but no objective cognitive deficits on a neuropsychological assessment [4]. The 2021 Behavioral Risk Factor Surveillance System reported that 12.67% of US adults aged 45 years and over experienced SCD [5]. Across the world (U.S., Australia, the Netherlands, Spain, Germany, Europe, Greece, France), an estimated 54% SCD prevalence was reported on a multicenter community-based and memory clinic study among adults aged 58–82 years [6]. SCD may be more prevalent with lower levels of activities in daily living [2,7]. For example, older adults with difficulties in social skills, which is a sub-domain of IADL, endorse higher cognitive decline [8]. Moreover, subjective perception of a reduction in function may contribute to or be associated with depressive symptoms from frequent irritations with lapses in daily functioning [2,9], increasing the burden of care.

Subjective cognitive decline may co-occur with depressive symptoms. For instance, people with a history of depression are likely to self-report subjective functional difficulties (SFD) [2,7]. A critical symptom of depression among older adults is a sense of disconnectedness from routine daily activities and a sense of frequent mentally unhealthy days (MUDs) [10], a reliable daily stressor indicator [11,12]. Despite the expected relationships between subjective cognitive decline and MUDs [1,2,13], their association is less well studied in older adults by age cohort. While a previous study examined SCD in instrumental activities of daily living (IADLs) [13], the literature on the role of MUDs in the relationship between subjective functional difficulties and SCD among older adults is lacking.

Specifically, the mechanism underlying the relationship between subjective cognitive decline and the frequency of MUDs remains unclear. MUDs may be a significant mediator and the age cohort a moderator.

### 1.1. Chronic Stress at an Older Age

The chronic stress theory of aging suggests that an individual’s psychological reactivity toward daily adverse events may determine their mental well-being [14,15] (Almeida et al., 2005; 2009). These minor daily stressors may trigger the affective responses associated with general affective distress [11,12,16] and the experience of frequent mentally unhealthy days (MUDs) [12]. Older adults with SCD may experience reduced activities of daily living than was the case at a younger age, contributing to their chronic distress [14,15], which they may find emotionally distressing [15]. Appraisal of the stressors and the variability in positive or negative affect is less well studied in older adults self-reporting subjective cognitive decline and functional difficulties. Nevertheless, evidence suggests that the average levels of negative affect people experience in responding to minor daily events reflect the wear-and-tear stress response [11,16], increasing risks of cognitive decline changes [17]. While past studies have suggested that engagement in physical and social activities may buffer emotional distress [11,18], fewer studies have examined different levels of functional difficulties affecting psychological distress measured by MUDs.

### 1.2. Mentally Unhealthy Days and SCD

Older adults with more subjective functional difficulties are more likely to experience SCD [1,2,19], while the frequency of associated MUDs may vary depending on their life situations. For instance, the stressor–reactivity path [12] proposes that self-perceptions of environmental and contextual factors influence the affective response and that modifying the perception of daily stressors for greater coping is associated with higher well-being [11]. Older adults’ emotional distress from daily stressors may be higher due to their self-appraisal of frequent MUDs [11,12,16]. For example, a longitudinal study using the Medicare Health Outcome Survey from 2004 to 2006 found that older adults with multiple chronic diseases reported more MUDs when having more ADL limitations [20]. Therefore, we speculate that subjective functional difficulties may be indirectly associated with older adults’ SCD through the mediating effect of MUDs. Understanding the implications of aging wear and tear in those with cognitive and functional impairment may help plan and manage care for people with SCD.

### 1.3. Subjective Functional Difficulties and SCD

Subjective difficulties in basic activities of daily living (ADLs) may present as failures in significant life domains in older adults associated with early-onset dementia [2,7]. Basic ADLs include bathing, toileting, eating, and grooming [21,22]. Competence in these activities is presumed for instrumental activities of daily living (IADLs), which involve more advanced tasks, such as civic and community life, finances, or a healthy lifestyle [2,7]. A subjective decline in basic ADLs would impair IADLs without targeted interventions in older adults with SCD [1,2,23,24] For example, older adults with mild cognitive impairment reported being less capable in their IADLs [23,25], increasing the risk of conversion to cognitive impairment [2,7,19,26,27].

**Age cohort effects**. The relationship between subjective functional difficulties and SCD may differ by age cohort, where older adults face more difficulties as they age. For example, a prior study showed that disability in ADLs was 1.5 times likely in the 80–84 age cohort and twice likely in the ≥85 age cohort compared to the 75–79 age cohorts [28]. However, more recent findings are mixed on the functional difficulties and SCD by age cohort [2,7,27,29], given other confounders. Personal resources, such as adequate assistance with ADLs [30], access to a healthcare professional [13,19], and varied social activity [18], may vary the levels of functional disability in old age. For example, psychological factors, such as a history of depressive disorders, often compound SCD [2,7,27,29]. Understanding how subjective functional difficulties and SCD vary across age cohorts is under study. Our study aims to address this evidence gap. Findings on changes in subjective functional difficulties leading to noticeable cognitive complaints would be necessary for interventions to improve the quality of life for older adults.

**Sociodemographic differences**. Subjective cognitive decline in older adults may differ by sociodemographic factors. For example, older age, males, lower education and income level, and minority groups are associated with worse cognitive decline [19,31,32]. However, there are variations in their SCD to their circumstances. For example, a systematic study reviewing 88 study populations found the effects of education on SCD were steadier in developed countries than in developing countries when gender, age, and race were considered [33]. In the presence of functional limitations, the experience of SCD would be compounded by a lower socio-economic gradient [31]. Older age, males, and Blacks are associated with having more disrupted IADLs [34]. For instance, older men are 64% more likely to report SCD than their counterparts [35]. While carrying a higher risk for depression over the life span [36], women seem to have greater overall daily functionality than men [37]. Despite these associations, sociodemographic disparity seems to de-intensify as age increases [38]. Moreover, the complex interaction among these sociodemographic variables on SCD in older adults with subjective functional difficulties by mentally-unhealthy-day mediation and age cohort moderation is less well studied. This study includes gender, education, income levels, and race as covariates to address the gaps in the evidence.

### 1.4. This Study

We examined the cross-sectional association of subjective functional difficulties with SCD and the mediating role of MUDs in the association. We also examined the moderating role of the age cohort on the relationship between subjective functional difficulties and SCD. By convention, categorical sociodemographic variables are moderators, while experience continuum variables are mediators [39]. Following our conceptual model (see Figure 1), we hypothesized that:Mentally unhealthy days mediate the relationship between subjective aging difficulties and SCD, controlling a history of depressive disorders, so greater subjective aging difficulties and MUDs are associated with higher SCD.The age cohorts moderate the indirect (mediation) effect of MUDs between subjective aging difficulties and SCD, controlling a history of depressive disorders. The predictive effect is stronger in the 80-year-old and older-age cohorts.Subjective aging difficulties and SCD are higher along the older-age cohorts in males, racial minorities, and people with lower education and income levels.

Despite the expected associations between subjective aging difficulties and SCD, studies are yet to map the likely mediation of MUDs in SCD and subjective aging difficulties, as measured on ADL and IADL difficulties, among older adults aged 65 years and over. These relationships need further study in sufficiently powered research as with national databases. Findings will clarify the profiles of SCD among older adults with subjective aging difficulties across age cohorts, which is essential to determine appropriate interventions to improve their daily quality of life.

## 2. Materials and Methods

### 2.1. Design and Sample

We used the 2019 Behavioral Risk Factor Surveillance Survey (BRFSS) data from the Centers for Disease Control and Prevention [40], a large, nationally representative sample of 121,099 noninstitutionalized US adults aged 18 years or older. For this study, we analyzed data from 44,778 cases of individuals aged ≥65. We omitted 27,995 cases with missing data. We then treated responses such as “Don’t know/not sure” and “Refused” with mode imputation when they were lower than 10% for each category and omitted them when they were more than 10%, leaving 13,377 cases for analysis.

### 2.2. Sociodemographic Characteristics of the Participants

The 13,377 participants included in this study were older adults aged 65 and over. They comprised 5880 (44%) men and 7497 (56%) women. The age ranged from 65 to ≥80 years, which included four age cohorts: 65–69 years old (30.8%), 70–74 years old (27.7%), 75–79 years old (18.9%), and ≥80 years old (22.7%). Across the participants, a more significant portion of the participants completed college or technical school (37.5%), followed by 30.4% who graduated high school, 27.4% who attended college or technical school, and a tiny percentage (4.7%) who did not graduate high school. Most participants (42%) earned ≥50,000 USD, only 7.5% earned less than 15,000 USD, and the remaining made something in between. Across the participants, Whites dominated (92.7%), followed by 4.9% African Americans, 1.0% American Indian or Alaskan Native, 0.9% from other races, and 0.4% Asians.

### 2.3. Variables and Measurement

**Criterion variable**. SCD was assessed based on the self-report on two questions [40]: (1) During the past 12 months, have you experienced the confusion of memory loss happening more often, or is it getting worse? (2) Because of a physical, mental, or emotional condition, do you have serious difficulty concentrating, remembering, or making decisions? Both questions are dichotomous, with yes (coded 1) or no (coded 0) responses. A higher score indicates a higher level of SCD. In this study, Cronbach’s alpha for the SCD was fair (0.516).

**Predictor variables.** Subjective aging difficulties were assessed based on the self-report of three yes (coded 1)/no (coded 0) questions assessing difficulties performing basic and instrumental daily living activities (B-ADLs and IADLs) [40]: (1) Do you have serious difficulty walking or climbing stairs? (2) Do you have difficulty dressing or bathing? (3) Because of a physical, mental, or emotional condition, do you have difficulties doing errands alone such as visiting a doctor’s office or shopping? Higher scores represent a higher difficulty level. In this study, Cronbach’s alpha for the subjective aging difficulties (SFD) was fair (0.590).

**Mediator.** Mentally unhealthy days were assessed as the self-reported average number of days during the past 30 days on which an older adult’s mental health was not good [40]. It is a self-report of continuous data ranging from 0 to 30. A higher number of reported days represents a higher number of poor mental health days.

**Moderator.** In this study sample, the age of participants was subdivided into four cohorts: 65–69 (4115, 30.8%), 70–74 (3700, 27.7%), 75–79 (2523, 18.9%), and ≥80 (3039, 22.7%).

**Covariates**. This study assessed self-reported gender (males and females), education level (did not graduate high school, graduated high school, attended college or technical school, or graduated college or technical school), income level (<15,000, 15,000–<25,000, 25,000–<35,000, 35,000–<50,000, and ≥50,000 USD), and racial group (White, Black and African American, American Indian or Alaskan Native, Asian, Native Hawaiian or other Pacific Islander, other race, and no preferred race).

### 2.4. Ethical Standards

This BRFSS dataset is open access available from the CDC [40], which provides de-identified data for public use. Therefore, there is no requirement for Institutional Review Board approval by the author’s institution or affiliation for secondary data analysis studies that use publicly accessible de-identified data.

### 2.5. Data Analysis

Before regression analysis, we checked for variance inflation factor values. We observed variance inflation factor values of 1.01–1.25 and tolerance values of 0.84–0.99, indicating no multicollinearity and residual problem. As a result, all assumptions were met per Field’s guidelines [40].

We used multiple regression and the bias-corrected percentile Bootstrap method for the study. The theoretical model was tested by estimating the 95% CI for mediation and moderated mediation effects with 5000 bootstrap samples. The statistics were considered statistically significant if the 95% CI did not include 0 [39,41], indicating that subjective functional difficulties significantly, indirectly affected SCD through mentally unhealthy days (MUDs). For the moderation variable (age cohorts) effect, we divided older age into four levels, 65–69, 70–74, 75–79, and ≥80, following the guidelines in [42]. We then used split-plot methods [43] to further examine the moderation effect’s direction, and a diagram was used to explain the moderation effect [44]. We tested the mediation and moderated models with the PROCESS V.4.0 macro for SPSS. We used model 4 in SPSS [39] to test the mediating effect of MUDs on the relationship between subjective aging difficulties and subjective cognitive decline (SCD). We used model 59 to test the moderated mediation model while controlling for age, gender, education, income level, and race. All analyses were performed using IBM SPSS V.29.0 (IBM, Armonk, NY, USA).

## 3. Results

### 3.1. Bivariate Correlation of Main Variables

The mean, SD/median (quartile_1_, quartile_3_), and correlation coefficient of each variable are shown in Table 1. As expected, subjective cognitive decline was positively associated with subjective aging difficulties (r = 0.244, *p* < 0.001), MUDs (r = 0.267, *p* < 0.001), a history of depressive disorder (r = 0.232, *p* < 0.001), age cohort (r = 0.076, *p* < 0.001), and race (r = 0.020, *p* < 0.001) and negatively associated with income level (r = −0.134, *p* < 0.001) and education (r = −0.081, *p* < 0.001). Subjective functional difficulties were positively associated with MUDs (r = 0.210, *p* < 0.001), a history of depressive disorder (r = 0.173, *p* < 0.001), age cohort (r = 0.148, *p* < 0.001), race (r = −0.153, *p* < 0.001), and gender (r = 0.037, *p* < 0.001) and negatively associated with education (r = −0.248, *p* < 0.001) and income level (r = −0.250, *p* < 0.001). MUDs were positively associated with a history of depressive disorders (r = 0.386, *p* < 0.001) and gender (r = 0.053, *p* < 0.001); negatively associated with age cohort (r = −0.058, *p* < 0.001), education (r = −0.040, *p* < 0.001), and income level (r = −0.109, *p* < 0.001); and not statistically associated with race (r = 0.0118, *p* = 0.204).

### 3.2. Subjective-Functional-Difficulty Mediation of SCD by MUDs

As shown in Table 2 and Table 3, subjective functional difficulties (SFD) had a significant predictive effect on SCD (*β* = 0.121, *t* = 20.76, *p* < 0.001), meaning a greater difficulty with ADLs/IADLs corresponds to a higher sense of SCD. The direct predictive effect of SFD was still significant when the mediating variable of mentally unhealthy days (MUDs) was added. In the path from SFD to MUDs, subjective functional difficulties had a significant positive predictive effect on MUDs (*β* = 1.424, *t* = 18/80, *p* < 0.001), signifying that MUDs increase as SFD increases. In the path from MUDs to SCD, MUDs also had a significant positive predictive effect on SCD (*β* = 0.014, *t* = 21.26, *p* < 0.001), indicating that as MUDs increase, SCD also increases.

In addition, the upper and lower limits of the bootstrap 95% CI for the direct effect of subjective functional difficulties (SFD) on SCD and the mediating effect of MUDs on SFD and SCD did not include 0 (Table 3), indicating that the mediating effect was statistically significant. The mediating effect value was 0.051, and the 95% CI was (0.033 to 0.02), which accounted for 14.12% of the total effect. This value showed that MUDs partially mediate the relationship between subjective functional difficulties and SCD. Our first hypothesis was fully supported, such that mentally unhealthy days mediate the relationship between SFD and SCD, controlling age cohorts, a history of depressive disorders, gender, education, income level, and race. We also found that more SFD and greater MUDs are associated with higher SCD.

### 3.3. Age Cohort Moderation of the Mediation Effect of MUDs on the Relationship between Subjective Functional Difficulties and SCD

As shown in Table 4, dummy variables were created for the four age cohorts as a moderator, with age cohort 1 as a reference. Moderator 1 (W1) represented age cohort 2 versus 1, moderator 2 (W2) defined age cohort 3 versus 1, and moderator 3 (W3) represented age cohort 4 versus 1. After the age cohort was added to the model, the interaction term between subjective functional difficulties (SFD) and W2 (age cohort 3 versus 1; *β* = −0.691, *t* = −3.18, *p* < 0.001) and between SFD and W3 (age cohort 4 versus 1; *β* = −1.10, *t* = −5.68, *p* < 0.001) significantly predicted mentally unhealthy days (MUDs); see Table 4, model 1. Figure 2 shows a simple slope analysis indicating the statistically significant interaction at all age cohorts (65–69, 70–74, 75–79, and ≥80 years old), with the 65–69 age cohort having the most pronounced effect (*b_simple_* = 1.98, *t* = 13.83, *p* < 0.001).

In the path from mentally unhealthy days (MUDs) to SCD, the interaction term between MUDs and age cohorts was not statistically significant at all levels (Table 4, model 2), as demonstrated in Figure 3. In the direct path from SFD to SCD, the interaction term between SFD and W2 (age cohort 3 vs. 1; *β* = −0.035, *t* = −2.07, *p* < 0.038) and SFD and W3 (age cohort 4 vs. 1; *β* = −0.066, *t*= −4.35, *p* < 0.001) significantly predicted SCD (Table 4, model 2). Figure 4 shows that the predictive effect of SFD on SCD was statistically significant at all age cohorts, with the 65–69 age cohort having the most pronounced effect (*b_simple_* = 1.54, *t* = 13.74, *p* < 0.001). In addition, the mediating effect value of mentally unhealthy days (MUDs) on the relationship between subjective functional difficulties (SFD) and SCD was substantial at all levels of the age cohort (moderated mediation effect), with the most notable effect in the 65–69 age cohort (*b* = 0.025, 95% CI 0.017–0.034) (Table 5). These findings were not consistent with our second hypothesis.

Through analysis of variance (ANOVA), we found that SCD is significantly associated with age cohorts (*F*(6, 13,373) = 24.616, *p* < 0.001). Pairwise comparisons showed that the 65–69 age cohort had greater SCD than the 70–74 *(M_difference_=* 0.074 (95% 0.43, −0.105, *p* < 0.001), 75–79 (*M_difference_ =* 0.052 (95% −0.018, 0.86, *p* < 0.001), and ≥80 (*M_difference_ =* 0.098 (95% 0.068, 0.129, *p* < 0.001) age cohorts.

### 3.4. Sociodemographic Effects

We found that worse SCD is associated with lower education levels (*b* = −0.013, *t* = −2.80, *p* = 0.005), lower income levels (*b* = −0.013, *t* = −4.03, *p* < 0.001), and female older adults (*b* = −0.022, *t* = −2.71, *p* = 0.007). SCD was positively associated with race (*b* = 0.021, *t* = 3.19, *p* = 0.001). We followed these findings with post hoc ANOVA and found that SCD is significantly associated with race: *F*(6, 13,373) = 5.21, *p* < 0.001. Pairwise comparisons showed that Asians *(M_difference_ =* −0.329 (95% −0.576, −0.081, *p* = 0.002) have greater SCD than those in the “other race” category, followed by Whites (*M_difference_* = −0.212 (95% −0.347, −0.077, *p* < 0.001), Blacks or African Americans *(M_difference_* = −0.197 (95% −0.342, −0.051, *p* = 0.001).

## 4. Discussion

The association between subjective functional difficulties (SFD) and SCD has been confirmed in the literature. This study examined the association between SFD and SCD by mentally-unhealthy-day (MUD) mediation and age cohort moderation, controlling a history of depressive symptoms, gender, education, income, and race. Results indicated that greater subjective functional difficulties are associated with more frequent MUDs, leading to a higher sense of SCD. Age cohorts moderated the relationship between SFD and SCD and between SFD and SCD but not between MUDs and SCD. The predictive effects of the path from SFD to MUDs and from MUDs to SCD were more substantial in the younger-old (65–69) than in the middle-old (70–79) and the oldest-old (≥80) age cohorts. Being a female older adult with lower education or income levels was associated with higher SCD. Regarding racial differences, Asian had higher SCD than older adults in the “other race” category, followed by Whites and Blacks or African Americans.

### 4.1. Mentally-Unhealthy-Day Mediation

The findings indicated that mentally unhealthy days (MUDs) mediate the relationship between subjective functional difficulties (SFD) and SCD, compounding SCD so that MUDs increase as SFD increase. The partial mediation effect accounting for 14.12% of the variance may be small. Nonetheless, in this study, we controlled for sociodemographic (gender, education, income level, and race) and psychological (history of depressive disorder) variables affecting MUDs and SCD, limiting biased study findings. The partial mediation may be due to a wide range of functional difficulties not assessed in this study, suggesting assessment of other ADL/IADL indicators of SCD (i.e., difficulties with mobility, communication, community, social, and civic life) for future study. Nonetheless, frequent irritations with daily activities are associated with a higher sense of SCD, as evident in past studies on SCD and daily living activities [2,25].

The chronic stress theory of aging suggests that aversive daily stressors (i.e., increased difficulties with ADLs) may heighten individuals’ negative reactivity [14] (Almeida et al., 2005) associated with general psychological distress [11,12,16]. Confirming previous SCD mediation study findings (Komalasari et al., 2022b), our results also extend the wear-and-tear theory of aging, suggesting an individual’s accentuated emotional vulnerability from increased functional difficulties, compounding SCD. An increased predisposition to an inflammation response toward stress, a biological reaction to repeatedly adjusting to stressors [14,16], may underlie these findings.

### 4.2. Age Cohort Moderation between the Variables

Our study showed that despite the overall mediation effect of MUDs in the cross-sectional association of SFD and SCD, age cohorts do not moderate the relationship between MUDs and SCD, unlike prior findings [13]. There is a tendency toward a higher sense of SCD in the younger-old (65–69) than in the middle-old (70–79) and ≥80 age cohorts. These findings lined up with previous evidence that the younger-old (65–69) age cohort tends to self-report more difficulties with ADLs and chronic diseases (i.e., arthritis and chronic obstructive airway disease) [45]. Interestingly, we found that the oldest-old (≥80) age cohort had more frequent MUDs than the younger-old (65–69) age cohort. This finding is consistent with previous findings that the oldest-old (≥80) age cohort report more psychopathology with advancing age, given their less frequent negative emotions and more positive affect than their middle-old and younger-old counterparts [46,47]. Accumulated empathy [48], gratitude [49], or varied social activity [18] may buffer their negative emotions to decrease functional abilities.

### 4.3. Sociodemographic Associations

Our findings supported the hypothesis that higher SCD is associated with lower formal education and income levels, confirming prior results [19,31]. Unlike previous study findings [34], being a female older adult was associated with lower SCD in this study. Regarding racial differences, Asians had higher SCD than older adults in the “other race” category, followed by Whites and Blacks or African Americans, conflicting prior findings that African Americans have a higher risk for SCD than Whites [19,31,34].

### 4.4. Study Contributions and Implications

We extended the predictions of chronic stress theory that affective reactivity to a daily stressor (i.e., greater ADL difficulties) may heighten the experience of SCD. We controlled for comorbid depressive disorders to understand the psychological distress toward functional difficulties implicated in the wear-and-tear theory of aging [12,14,15].

SCD is self-perceived cognitive performance that implicates intact daily functioning independent of cognitive capabilities [4], such as daily functioning. Nevertheless, our findings contribute to a better understanding of the influence of emotional reactivity to daily stressors, such as subjective functional difficulties, on determining SCD. Although people with SCD might already have a subjective memory complaint (Studart et al., 2016), they might not show progressive cognitive performance on a formal objective neuropsychological assessment (Jessen et al., 2020). Nonetheless, the negative affect associated with daily living difficulties and self-perceived cognitive difficulties can lead to worry and anxiety out of fear of dementia (Smart et al., 2016). These findings add to the literature on the importance of observing older adults’ ADL/IADL difficulties, which may indicate an incipient cognitive decline. Person-centric intervention for older adults with SCD to self-monitor in ADLs may be indicated to improve their quality of life. Mindfulness training is one of the feasible early interventions effective for people with SCD to reduce cognitive complaints (Smart et al., 2016), countering the negative affect associated with difficulties performing daily living activities.

### 4.5. Limitations of the Study and Suggestions for Further Research

This study was self-reporting and correlational. Future studies could research variables through different methods to overcome the correlational nature of the study findings. For example, subjective functional difficulties can be measured on recorded tasks (i.e., making a phone call, preparing a pillbox in a specific order within 15 min) and observed on an actigraphy [50]. The use of cross-sectional data collection was a limitation, although it was carried out among a relatively large sample. A longitudinal research design would lend greater confidence to the findings. Although this study partially mediated the mental health of the relationship between subjective functional difficulties and SCD, there could be other mediating variables in this relationship for future research. Finally, the significant associations were weak, suggesting a need for further aging health and function studies using similar data sets as the BRFSS.

## 5. Conclusions

Our study examined the association between subjective functional difficulties and SCD and the potential mediating role of mentally unhealthy days (MUDs) in this relationship in a national US sample of older adults 65 years old and over. We supported our first hypothesis that MUDs mediate the relationship between SFD and SCD. The second hypothesis was not supported. We found that the mediation effect of SFD on SCD by MUDs is more pronounced in the younger-old (65–69) age cohort than in the older-old (70–74) and ≥80 age cohorts. Worse SCD was associated with being Asian, in female older adults, and in those with lower education and income levels. These findings extend the previous predictions of chronic stress aging, confirming the potential impact of functional difficulty accentuating emotional sensitivity, contributing to higher SCD in the younger-old age group. Targeted person-centric mindfulness interventions by age cohort through self-monitoring in ADLs may slow SCD progression to a less severe level in older adults with psychological distress. Practitioners and caregivers of older adults may benefit from understanding the role of MUDs in mediating the relationship between SFD and SCD.

## Figures and Tables

**Figure 1 ijerph-20-01606-f001:**
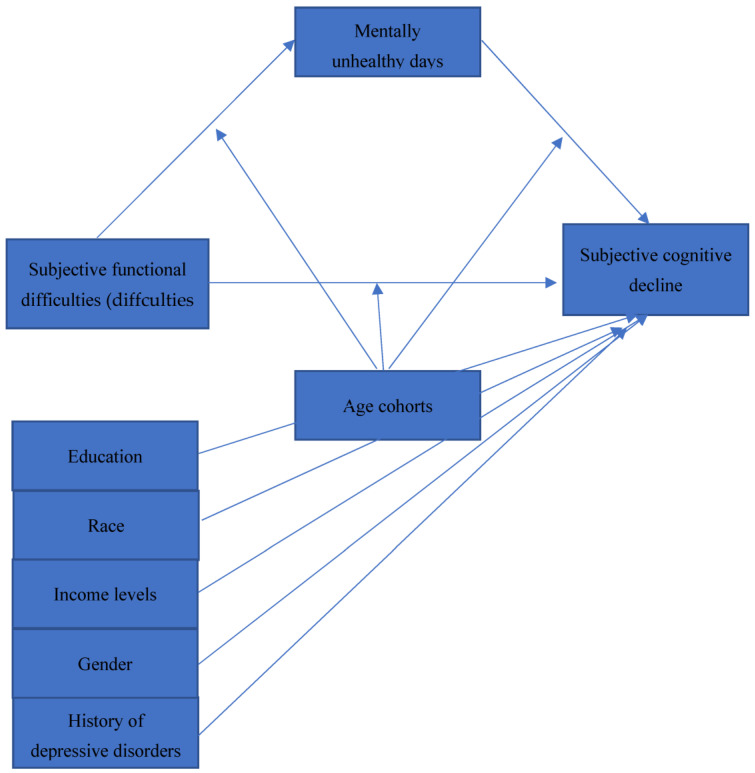
The conceptual framework of the moderated mediation model of the effect of mentally unhealthy days on SFD and SCD, controlling a history of depressive disorders, gender, education, income levels, and race. SCD, subjective cognitive decline; SFD, subjective functional difficulties).

**Figure 2 ijerph-20-01606-f002:**
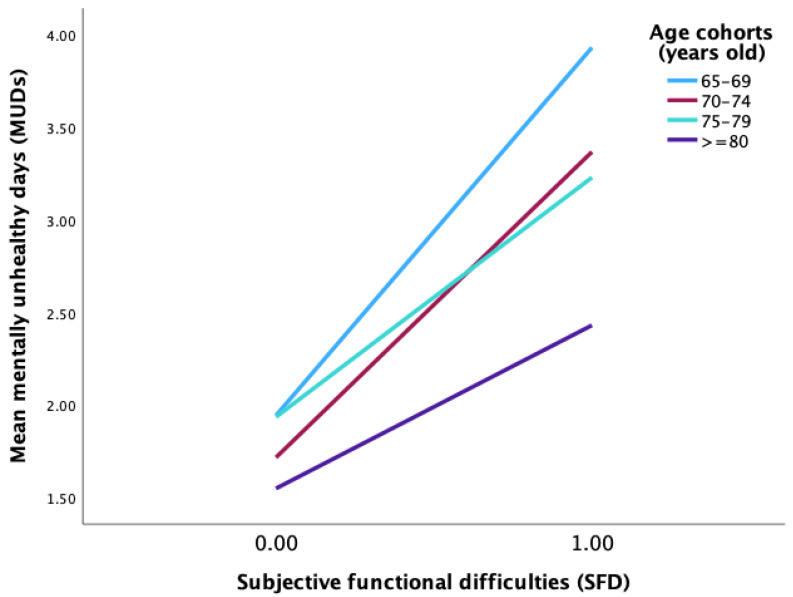
Simple slope analysis showed that the predictive effect of SFD on MUDs was statistically significant in all age cohorts, with the strongest effect in the younger-old (65–69) age cohort.

**Figure 3 ijerph-20-01606-f003:**
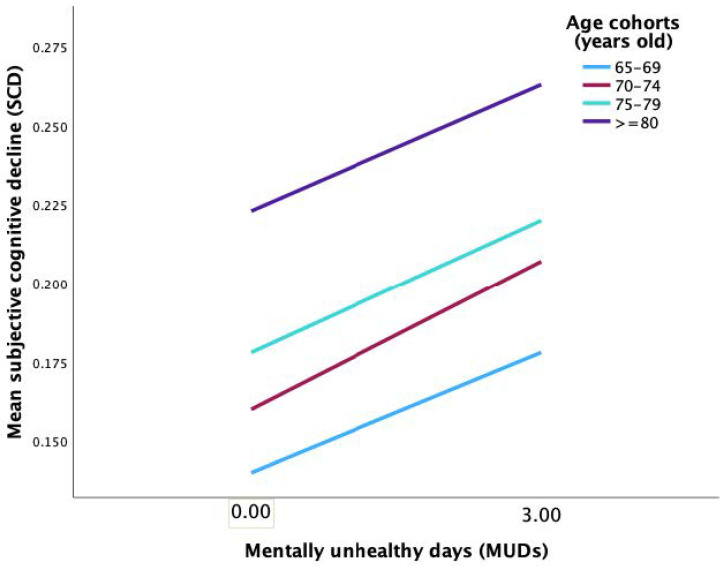
Simple slope analysis showed that the predictive effect of mentally unhealthy days on the SCD was not statistically significant at each level of the age cohorts.

**Figure 4 ijerph-20-01606-f004:**
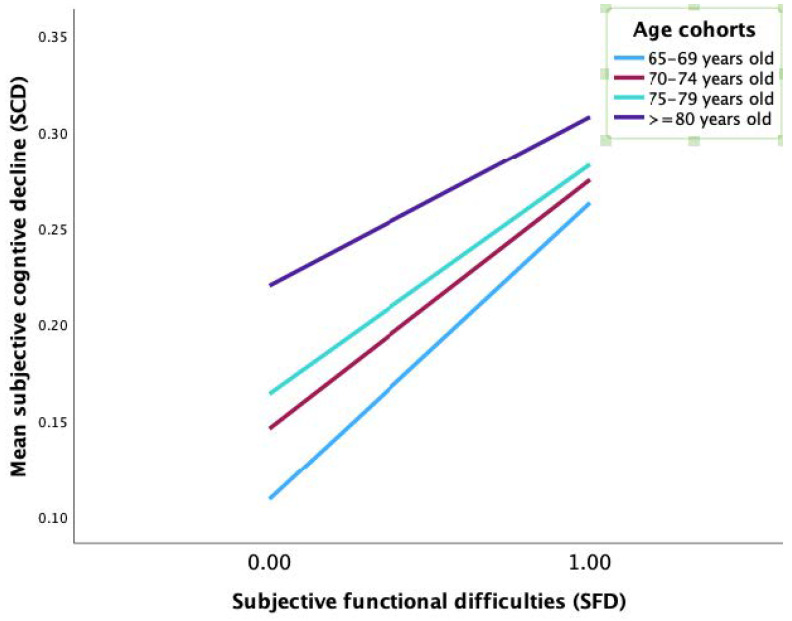
Simple slope analysis showed that the predictive effect of SFD on SCD was statistically significant in all age cohorts, with the strongest effect in the younger-old (65–69) age cohort.

**Table 1 ijerph-20-01606-t001:** Spearman correlation coefficients of variables among participants.

Variable	M ± SD	M ± (P_25,27_)	1	2	3	4	5	6	7	8	9
1. SCD	0.202 ± 0.49		1								
2. SFD	0.393 ± 0.722		0.244 *	1							
3. MUDs	2.34 ± 6.55		0.267 *	0.210 *	1						
4. History of depression		0 (0,0)	0.232 *	0.173 *	0.3868	1					
5. Education		3 (2,4)	0.148 *	−0.153 *	−0.040 *	−0.017 ***	1				
6. Race		1 (1,1)			0.011 ***	−0.025 **	−0.042 *	1			
7. Income level		4 (2,5)	0.065 *	−0.250 *	−0.109 *	−0.094 *	0.451 *	−0.084 *	1		
8. Gender		2 (1,2)			0.053 *	0.045 *	−0.040 *	0.013 ***	−0.071 *	1	
9. Age cohort			0.076 *	0.037 *	−0.058 *	−0.085 *	−0.102 *	−0.013 ***	−0.166 *	0.018 ***	1

* *p* < 0.001; ** *p <* 0.05; *** *p* ≥ 0.05. M ± SD: mean ± standard deviation; M ± (P_25,27_): median (quartile, quartile). 1,2,3 follows a normal distribution and is described as M ± SD. 4,5 do not follow a normal distribution and are described as M ± (P_25,27_). History of depressive symptoms: 0 (no); 1 (yes). Education levels: (1) did not graduate high school, (2) graduated high school, (3) attended college or technical school, and (4) graduated college or technical school. Income levels: (1) <15,000, (2) 15,000–<25,000, (3) 25,000–<35,000, (4) 35,000–<50,000, and (5) ≥50,000 USD. Race: (1) White, (2) Black and African American, (3) American Indian or Alaskan native, (4) Asian, (5) Native Hawaiian or other Pacific Islander, (6) other race, and (7) no preferred race. Gender: (1) male and (2) female. SCD, subjective cognitive decline; SFD, subjective functional difficulties; MUDs, mentally unhealthy days.

**Table 2 ijerph-20-01606-t002:** Testing the mediation effect of mentally unhealthy days (MUDs) between subjective functional difficulties and SCD in the U.S., 2019.

Predictors	Model 1 (SCD, Total Effect)				Model 2 (MUDs)				Model 3 (SCD)			
	** *β* **	** *t* **	**95% CI**		** *β* **	** *t* **	**95% CI**		** *β* **	** *t* **	**95% CI**	
			LL	UL			LL	UL			LL	UL
SFD	0.141	24.08 *	0.129	0.152	0.121	20.75 *	0.110	0.132	0.121	20.75 *	0.110	0.132
MUDs	-	-	-	-	-	-	-	-	0.014	21.75 *	0.110	0.132
History of depression	0.292	24.78 *	0.269	0.315	6.05	39.71 *	5.75	6.35	0.207	16.92 *	0.183	0.231
Age cohort	0.023	0.624 *	0.016	0.030	−0.258	−5.51 *	−0.035	−0.166	0.026	7.35 *	0.019	0.033
Education	−0.016	−3.25 **	−0.025	−0.006	−0.016	−5.51 *	−0.350	−0.166	−0.014	−2.85 **	−0.023	−0.004
Race	0.022	3.36 **	0.009	0.036	0.095	0.099 ***	−0.074	0.264	0.021	3.21 **	0.008	0.034
Income level	−0.018	−5.21 *	−0.024	−0.011	−0.244	−5.59 *	−0.330	−0.264	−0.014	−4.27 *	−0.021	−0.008
Gender	0.021	−2.55 **	−0.037	−0.005	0.105	1.00 ***	−0.100	0.312 center	−0.022	−2.78 ***	−0.038	0.0007
*R* ^2^	0.120				0.162				0.149			
*F*	260.39				370.03				292.01			

* *p* < 0.001; ** *p <* 0.05; *** *p >* 0.05. SFD, subjective functional difficulties; SCD, subjective cognitive decline.

**Table 3 ijerph-20-01606-t003:** The total effect, direct effect, and mediation effect of mentally unhealthy days on subjective aging difficulties and SCD.

Total Effect	*β*	SE	BootCI		
			LL	UL	
Direct effect	0.121	0.006	0.120	0.132	85.87%
Indirect effect	0.020	0.002	0.016	0.024	14.12%

BootSE, bootstrap standard error; BootCI, bootstrap 95% confidence interval. LL; lower level; UL: upper level.

**Table 4 ijerph-20-01606-t004:** Testing the moderated mediation effect of MUDs on SCD among older adults in the U.S., 2019.

Predictors	Model 1 (MUDs)				Model 2 (SCD)			
	*β*	*t*	95% CI		*β*	*t*	95% CI	
			LL	UL			LL	UL
SFD	1.98	13.83 *	1.70	2.26	0.154	13.74 *	0.103	0.194
MUDs	-	-	-	-	0.013	11.80 *	0.011	0.015
Age cohort 2 vs. 1 (W1)	−0.227	−1.51 ***	−0.522	0.068	0.031	2.66 **	0.008	0.054
Age cohort 3 vs. 1 (W2)	−0.009	−0.051 ***	−0.345	0.328	0.053	3.94 *	0.027	0.079
Age cohort 4 vs. 1 (W3)	−0.394	−2.31 **	−0.728	−0.060	0.109	8.29 *	0.083	0.135
SFD × W1	−0.336	−0.166 ***	−0.732	0.061	−0.026	−1.64 ***	−0.057	0.005
SFD × W2	−0.691	−3.17 **	−1.12	−0.265	−0.035	−2.07 **	−0.069	−0.002
SFD × W3	−1.10	−5.68 *	−1.48	−0.7.2	−0.066	−4.35 *	−0.096	−0.036
MUDs × W1	-	-	-	-	0.003	1.73 ***	0.000	0.006
MUDs × W2	-	-	-	-	0.001	0.621 ***	−0.002	0.005
MUDs × W3	1	-	-	-	0.001	0.333 ***	−0.003	0.004
History of depression	6.00	39.40 *	5.71	6.30	0.205	16.72 *	0.181	0.229
Education	−0.143	−2.30 **	−0.265	−0.021	−0.013	−2.80 **	−0.023	−0.004
Race	0.084	0.976 ***	−0.085	0.253	0.021	3.19 **	0.008	0.034
Income level	−0.233	−5.34 *	−0.319	−0.147	−0.013	−4.03 *	−0.020	−0.007
Gender	0.106	1.01 ***	−0.099	0.311	−0.022	−2.71 **	−0.037	−0.006
*R* ^2^	0.165				0.150			
*F*	219.98 *				147.68 *			

* *p* < 0.001; ** *p <* 0.05; *** *p >* 0.05. W1 (age cohort 2 vs. 1). W2 (age cohort 3 vs. 1), W3 (age cohort 4 vs. 1). SCD, subjective cognitive decline; SFD, subjective functional difficulties; LL, lower-level confidence interval; UL, upper-level confidence interval.

**Table 5 ijerph-20-01606-t005:** Mediating effect values at different levels of age cohorts among older adults in the U.S., 2019.

Age Cohort	Effect	BootSE	BootLLCI	BootULCI
65–69	0.154	0.011	0.018	0.034
70–74	0.128	0.012	0.106	0.151
75–79	0.119	0.013	0.094	0.145
≥80	0.088	0.010	0.068	0.109

BootSE, bootstrap standard error; BootLLCI, bootstrap lower-level confidence interval.

## Data Availability

The corresponding author’s data supporting this study’s findings are available upon request.

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
