# Peer review of "Subjective Functional Difficulties and Subjective Cognitive Decline in Older-Age Adults: Moderation by Age Cohorts and Mediation by Mentally Unhealthy Days"

_ijerph, 2023, doi:10.3390/ijerph20021606_

Round 1

Reviewer 1 Report

The current manuscript aims to explore the relationship between subjective functional and cognitive difficulties among older aduts, as well as the potential mediation role of mentally unhealthy days, age cohorts and a series of clinical and demographic variables.

The paper is well written: especially the introduction section appears to be comprehensive, and successfully covers the main literature on the topic. Some paragraphs are maybe too detailed, but overall the paper seems easy to read.

I only have a few minor issues to point out, mainly related to study design.

1. The study focuses on "subjective impairments" and on subjective reports by study participants. However, the presence of objective impairments has not been excluded, so we don't know whether people included in the study actually had concrete impairments in cognitive function or daily abilities. It would be better to specify whether this was an exclusion criterion for the study, or to check whether correcting for this parameter changes anything in study results. This could be true especially for the oldest-old age cohort.

2. A substantial proportion of cases were excluded from the analyses due to missing data and due to more than 10% of "Don't know/not sure" answers. This issue could partially relate to the previous one. Were these subjects somehow different from included subjects? The "Don't know/not sure" answers could also be related to subjective cognitive difficulties, and study variables (especially mentally unhealthy days) are highly subjective.

3. The fact that study variables (especially mentally unhealthy days) are highly subjective could be commented more explicitly.

4. The Authors explore the relationship between subjective cognitive and functional difficulties assuming that the former are determined by the latter. Could it be possibile to hypothesize that subjective functional impairment could derive from cognitive claims? This possibility has not been explored.

Author Response

Thank you for the valuable review that has improved our manuscript.

Reviewer's Comments

Authors' response

Reviewer #1

The current manuscript aims to explore the relationship between subjective functional and cognitive difficulties among older adults, as well as the potential mediation role of mentally unhealthy days, age cohorts and a series of clinical and demographic variables.

The paper is well written: especially the introduction section appears to be comprehensive, and successfully covers the main literature on the topic. Some paragraphs are maybe too detailed, but overall the paper seems easy to read.

I only have a few minor issues to point out, mainly related to study design.

1. The study focuses on "subjective impairments" and on subjective reports by study participants. However, the presence of objective impairments has not been excluded, so we don't know whether people included in the study actually had concrete impairments in cognitive function or daily abilities. It would be better to specify whether this was an exclusion criterion for the study, or to check whether correcting for this parameter changes anything in study results. This could be true especially for the oldest-old age cohort.

We appreciate the reviewer's positive feedback in summarizing this paper.

We thank the reviewer’s for commenting on the SCD criteria. True, we did not include objective data on the subjective cognitive decline nor used it as an exclusion criteria. The reason being a person with SCD might already have a subjective memory complaint (Studart et al., 2016), but not show a progressive cognitive performance on a formal objective neuropsychological assessment (Jessen et al., 2020). We added these lines to the manuscript to clarify this, as appeared in ‘Introduction’ lines 33-35 in page 1.

2. A substantial proportion of cases were excluded from the analyses due to missing data and due to more than 10% of "Don't know/not sure" answers. This issue could partially relate to the previous one. Were these subjects somehow different from included subjects? The "Don't know/not sure" answers could also be related to subjective cognitive difficulties, and study variables (especially mentally unhealthy days) are highly subjective.

We appreciate the reviewer’s careful observation on cases’ responses ‘Don't know/not sure’ or ‘Refused’ in our data. It is true that these subjects were excluded from analysis. As indicated in “Design and sample,” in line 195-198 in page 5 when such responses were less than 10% for a variable, we conducted mode imputation but when they were more than more 10%, we omitted these responses, which should not bias the study results.

3. The fact that study variables (especially mentally unhealthy days) are highly subjective could be commented more explicitly.

4. The Authors explore the relationship between subjective cognitive and functional difficulties assuming that the former are determined by the latter. Could it be possibile to hypothesize that subjective functional impairment could derive from cognitive claims? This possibility has not been explored.

We appreciate the reviewer’s suggestion to emphasize subjectivity of variables assessed in this study. We have added ‘self-report’ to clarify subjective nature of the study variables, as appeared in line 226 in page 5 and line 233 in page 6.

We appreciate the reviewer’s critical observation on the possibility that subjective functional difficulties occur due to subjective cognitive decline. While the results of this study show that greater subjective functional decline is associated with higher SCD, it should be noted that SCD is a pre-mild cognitive impairment (MCI), a stage implicates an intact daily functioning (Peterson et al., 2004). To clarify why we did not explore this possibility, we have added the following in the “Study contributions and implications”: “SCD is a self-perceived cognitive performance which implicates an intact daily functioning (Peterson et al., 2004) independent of cognitive capabilities (Jessen et al., 2014), such as daily functioning. Nevertheless, our findings added to the literature on the importance of observing older adults’ ADL/IADL difficulties, which may indicate an incipient cognitive decline,” as appeared in line 423-429 in page 14. We also cited a reference (Jessen et al., 2014) to support this.

Reviewer 2 Report

Thanks for the opportunity to review this interesting paper. The current study examined the mediation of 11 MUDs of the association between SFD and SCD by age cohorts' moderation among older adults. The results indicated that The bias-corrected percentile bootstrap with 5000 samplings revealed that MUDs partially mediated the relationship between SFD and SCD (14.12% mediation effect), controlling depressive symptoms, education, income levels, and race. Age cohort moderated the relationship between SFD and SCD, SFD and SCD, but not MUDs and SCD. The predictive effects of SFD to MUDs and MUDs to SCD were stronger in the younger-old (65-69) than the middle-old (70-79) and oldest-old (80) age cohorts. Worse SCD was associated with being Asians, female older adults, and at lower education years and income levels.  This study is well done. I only have a few minor comments.

1. In the introduction, please report the international epidemiology statistics of SCD, not only the US.

2. In the discussion section, please only highlight the results, and compare the results with previous studies.

Author Response

We thank you for valuable review that helps improve our manuscript.

Reviewer's Comments

Authors' response

Reviewer #2

Thanks for the opportunity to review this interesting paper. The current study examined the mediation of 11 MUDs of the association between SFD and SCD by age cohorts' moderation among older adults. The results indicated that the bias-corrected percentile bootstrap with 5000 samplings revealed that MUDs partially mediated the relationship between SFD and SCD (14.12% mediation effect), controlling depressive symptoms, education, income levels, and race. Age cohort moderated the relationship between SFD and SCD, SFD and SCD, but not MUDs and SCD. The predictive effects of SFD to MUDs and MUDs to SCD were stronger in the younger-old (65-69) than the middle-old (70-79) and oldest-old (80) age cohorts. Worse SCD was associated with being Asians, female older adults, and at lower education years and income levels.  This study is well done. I only have a few minor comments.

1.     In the introduction, please report the international epidemiology statistics of SCD, not only the US.

2. In the discussion section, please only highlight the results, and compare the results with previous studies.

We appreciate the reviewer's positive feedback in summarizing this paper.

We appreciate the reviewer’s suggestion to add on report of SCD prevalence across the world. We have followed the suggestion and added “Across the world (USA, Australia, the Netherlands, Spain, Germany, Europe, Greece, France), an estimated 54% SCD prevalence was reported on a multicenter community-based and memory clinic study among adults aged 58-82 [6], as appeared on line 35-38 in page 1.

We thank the reviewer’s advice to limit the ‘Discussion’ of our results by comparing them to prior findings. We have now removed some information from ‘Discussion,’ as follows:

From line 3984-392 in page 13: The chronic stress theory of aging suggests that aversive daily stressors (i.e., increased difficulties with ADLs) may heighten individuals’ negative reactivity (Almeida et al., 2005) associated with general psychological distress (Charles et al., 2013; Hanh et al., 2014; Sin et al., 2015). Confirming previous SCD mediation study findings (Komalasari et al., 2022b), our present results also extended the wear and theory of aging, suggesting an individual’s accentuated emotional vulnerability from increased functional difficulties, compounding SCD. An increased predisposition to inflammation response towards stress, a biological reaction to repeatedly adjusting to stressors (Almeida et al., 2005; Sin et al., 2015), may underlie these findings.

From line 413-414 in page 13: Our study extends the evidence from previous SCD studies as we assessed difficulty concentrating, remembering, and making decisions compounded with frequent memory loss.

From line 420-424 in page 14: The findings of this study contribute to a better understanding of the influence of emotional reactivity to daily stressors like subjective functional difficulties on determining SCD. Although people with SCD might already have a subjective memory complaint (Studart et al., 2016), they might not show a progressive cognitive performance on a formal objective neuropsychological assessment (Jessen et al., 2020).

From line 427-4232 in page 15: Mindfulness training is one of the feasible early interventions effective for people with SCD to reduce cognitive complaints (Smart et al., 2016). The negative affect associated with daily living difficulties and self-perceived cognitive difficulties can lead to worry and anxiety out of fear of dementia (Smart et al., 2016). Mindfulness can help people with SCD to recognize and counter negative affect associated with difficulties performing daily living activities.”

We also removed references cited accordingly.